# Effect of Precipitation-Free Zone on Fatigue Properties in Al-7.02Mg-1.98Zn Alloys: Crystal Plasticity Finite Element Analysis

**DOI:** 10.3390/ma17225623

**Published:** 2024-11-18

**Authors:** Xin Chen, Xiaoyu Zheng, Meichen Pan, Yuling Liu, Yi Kong, Alexander Hartmaier, Liya Li, Yong Du

**Affiliations:** 1State Key Laboratory of Powder Metallurgy, Central South University, Changsha 410083, China; 213307007@csu.edu.cn (X.C.); xiaoyu_zheng@csu.edu.cn (X.Z.); panmeichen18012105@163.com (M.P.); liu.yuling@csu.edu.cn (Y.L.); yikong@csu.edu.cn (Y.K.); 2Interdisciplinary Centre for Advanced Materials Simulation (ICAMS), Ruhr-Universität Bochum, Universitätsstr. 150, 44801 Bochum, Germany; alexander.hartmaier@ruhr-uni-bochum.de

**Keywords:** Al-Mg-Zn, precipitation-free zone, crystal plasticity finite element method, crystal orientation, fatigue crack initiation

## Abstract

Age-strengthened aluminum alloys, as important lightweight structural materials, have significantly lower fatigue properties compared to non-age-strengthened aluminum alloys. In this study, the polycrystalline models containing precipitation-free zones (PFZ) were constructed by secondary development of the traditional polycrystalline model by modifying the mesh file. Polycrystalline finite element simulations of peak age-treated Al-7.02Mg-1.98Zn alloys were carried out with this model. The results demonstrate that the PFZ’s presence markedly reduces the alloy’s yield strength and a substantial stress concentration occurs adjacent to the PFZ, generating significant compressive stresses at the PFZ. Under cyclic loading, the maximum strain energy dissipation in the model containing the PFZ far exceeds that observed in the conventional polycrystalline model, and the strain energy dissipation observed in the PFZ is significantly higher than that at other locations. This indicates that the PFZ is the main region for fatigue crack initiation. In addition, the introduction of a rotation factor to simulate the inhomogeneous rotation within the grain reveals that the additional stress concentration in the PFZ introduced by the aluminum alloy-forming process further increases the fatigue crack initiation driving force.

## 1. Introduction

In recent years, issues related to energy shortages and the greenhouse effect have become increasingly severe, leading to a consensus on the necessity of promoting energy conservation and emission reduction through the lightweighting of structural components [1]. Aluminum alloys are among the preferred materials for lightweight structural applications because of their high specific strength, excellent corrosion resistance, favorable formability, and good recyclability [2,3,4]. Notably, 5XXX series alloys offer advantages such as low density, outstanding corrosion resistance, and good weldability [5,6]; however, their mechanical properties can be enhanced only through strain hardening and solid solution strengthening, resulting in significantly lower strength compared to 2XXX and 7XXX series alloys [7,8,9]. Recently, researchers have developed a new type of Al-Mg-Zn alloy that has a composition range between the conventional 5XXX and 7XXX aluminum alloys. By adding Zn to the conventional 5XXX aluminum alloy, this new alloy can precipitate a large number of fine and dispersed nano T-Mg_32_(Al,Zn)_49_ phases by artificial aging, while maintaining the excellent corrosion resistance and good weldability characteristic of 5XXX aluminum alloys. The precipitated phase significantly enhances the strength of the alloy through aging, resulting in improved mechanical properties and promising potential for applications in aerospace, transportation, and other fields [10,11,12].

In practical applications, such as aircraft frames and structural components, materials are subjected to alternating loads during service, which can lead to fatigue damage. Fatigue fracture is widely recognized as one of the primary failure modes in engineering structures. Due to the difficulty of detecting fatigue cracks in aluminum alloy components during service, implementing preventive measures becomes challenging, potentially leading to catastrophic consequences. Therefore, it is essential to understand the fatigue characteristics of aluminum alloys. Although the fatigue of metals has been studied for over 2 centuries, significant research is still required to understand and predict the conditions that lead to crack initiation in polycrystalline microstructures under cyclic loading. Fatigue crack initiation (FCI) and small crack propagation (SCG) are the two stages of crack development most influenced by the microstructure [13]. However, the results of fatigue experiments often exhibit substantial dispersion, resulting in considerable uncertainty in fatigue model predictions. Therefore, there is an urgent need to develop microstructure-based models to enhance the reliability of fatigue life calculations for engineering components [14].

The fatigue strength (a dynamic property) and tensile strength (a static property) are strongly correlated in the case of steels: fatigue strength/tensile strength~1/2. For non-age-strengthened aluminum alloys, the ratio is also about 1/2. However, the fatigue properties of age-strengthened Al alloys are relatively inferior, with their fatigue strength being ~1/3 of their tensile strength [15]. Cyclic loading leads to microplasticity and the accumulation of irreversible damage in the form of localization of plasticity (usually associated with defects). Plastic localization catalyzes FCI. A key feature of precipitation-strengthened aluminum alloys is the presence of a precipitation-free zone (PFZ) adjacent to grain boundaries. The PFZ is often considered to be a weak region in aluminum alloys because it lacks reinforcement from precipitation phases and exhibits lower strength compared to the intragranular region [16]. During fatigue loading, plastic deformation becomes localized in these soft deformation zones (PFZ), and the irreversible cyclic dislocation motion leads to damage accumulation, thereby creating conditions conducive to fatigue cracking.

Crystal plasticity finite element modeling simulations have been shown to model FCI accurately and efficiently [14]. To investigate the influence of microstructure on FCI, a computable fatigue indication parameter (FIP) can effectively characterize the shear strain-dominated fatigue crack driving force from a microscopic perspective [17]. Manonukul and Dunne [17] proposed a crack initiation criterion based on a critical accumulated plastic slip, where crack initiation occurs when the accumulated plastic slip reaches a critical value. Cruzado et al. [18] predicted the fatigue crack initiation (FCI) life by utilizing the strain energy dissipation in each fatigue load cycle and accurately estimated the location of fatigue crack initiation based on the energy dissipated due to slip, which provides a more explicit physical basis. Korsunsky et al. [19] and Skelton et al. [20] demonstrate the dependence of crack initiation and growth on local energy dissipation. Particularly at higher strain amplitudes, utilizing strain energy dissipation as the FIP can provide better predictions for metal low-cycle fatigue (LCF) that are consistent with LCF test results [21].

In this work, we construct a polycrystalline model that incorporates PFZ based on the microscopic properties of age-strengthened aluminum alloys. Strain energy dissipation is integrated as a fatigue-indicating parameter within the framework of the crystal plasticity model. The stress-strain response and local plastic deformation behavior under fatigue loading are simulated using the finite element method. The aim of this study is to explore the intrinsic relationship between PFZ and FCI life of aluminum alloys, as well as to examine the fatigue characteristics of age-strengthened aluminum alloys. This research provides a theoretical basis for enhancing the fatigue life of high-strength aluminum alloys, extending their application to a wider range of scenarios, and facilitating the lightweighting of structural components.

## 2. Materials and Methods

### 2.1. Three Types of Polycrystalline Geometry Models

This work investigates an age-strengthened Al-Mg-Zn alloy (Beijing, China, Al-7.02Mg-1.98Zn) [22]. The alloy underwent an artificial aging treatment of 90 °C for 24 h followed by 140 °C for 24 h to achieve the peak aging state, during which the alloy exhibits the highest strength. In this peak aging state, the alloy demonstrates a pronounced PFZ [7,8,23,24].

Age-strengthened aluminum alloys typically undergo deformation processes, such as extrusion and rolling, prior to solid solution aging. During the deformation of polycrystalline materials, grains are internally rotated homogeneously to maintain geometrical compatibility, influenced by the constraints imposed by neighboring grains [25,26,27].

Based on this, three distinct polycrystalline geometrical models will be developed in this study as representative volume elements (RVE) for numerical simulations. This will be achieved using the open-source software Neper (Version 4.9.0) and a secondary processing method [28]. The models include (A) a polycrystalline model without PFZ; (B) a polycrystalline model with one complete PFZ per grain; and (C) a model that separates the PFZ into multiple regions based on neighboring grains. Initially, a geometrical model containing 80 equiaxed grains was created using Neper’s Voronoi method and subdivided into 40,000 CPE4 quadrilateral grids (Model A). By traversing this set of elements to determine their boundary conditions and the adjacency of neighboring grains, Models B and C (local enlargement) were subsequently generated, as illustrated in Figure 1.

In this study, the PFZ orientation of Model B is aligned with the grain orientation. In Model C, the constraints due to neighboring grains during the deformation process lead to uneven grain rotation. A rotational path is constructed between two neighboring grains to represent the orientation difference between them. Depending on the degree of deformation, a different rotation factor (RF) is defined, allowing the PFZ orientation to be expressed as the product of the rotational orientation from the current grain to the neighboring grain and the corresponding RF. Figure 2 presents a schematic diagram illustrating the orientation of the PFZ in relation to the two neighboring grains. The green orientation represents the grain to which the PFZ belongs, while the black orientation indicates the neighboring grain. The small circle between the two larger circles represents the orientation corresponding to different rotation factors, which are set at 0.125, 0.25, 0.375, and 0.5.

### 2.2. Material Modeling

The crystal plasticity model is based on the work of Huang [29]. In this classical framework, the deformation gradient F is decomposed into the plastic part $F^{p}$ caused by crystal slip and the elastic part $F^{e}$ caused by elastic lattice deformation. The equation is as follows:(1)$$F=F^{e}\cdot F^{p}$$

Assuming plastic deformation occurs due to only the crystalline slip. The plastic velocity gradient is obtained based on the following relationship:(2)$$L^{p}={\dot{F}}^{p}\cdot {\left(F^{p}\right)}^{-1}=\sum_{\alpha =1}^{N}{\dot{\gamma }}^{\alpha }\left(m^{\alpha }⊗n^{\alpha }\right)$$
where the plastic slip rate $\dot{\gamma }$ is integrated over all slip systems, where ${\dot{\gamma }}^{\alpha }$ represents the slip rate of the α slip system, $m^{\alpha }$ and $n^{\alpha }$ denote the slip direction and the normal to the slip plane of the slip system, respectively.

The resolved shear stress $\tau^{\alpha }$ is given by:(3)$$\tau^{\alpha }=\sigma ∶{\left(m^{\alpha }⊗n^{\alpha }\right)}_{sym}$$

σ is the Cauchy stress tensor, sym represents the symmetric part of the tensor.

The slip rate ${\dot{\gamma }}^{\alpha }$ is determined according to the power law. In order to characterize the isotropic and kinematic hardening of the material under cyclic loading, the back stress $\chi^{\alpha }$ was introduced:(4)$${\dot{\gamma }}^{\alpha }={\dot{\gamma }}_{0}{\left|\frac{\tau^{\alpha }-\chi^{\alpha }}{g^{\alpha }}\right|}^{n}sign\left(\tau^{\alpha }-\chi^{\alpha }\right)$$
where ${\dot{\gamma }}_{0}$ is the reference strain rate of the slip system, n is the rate sensitivity index, and $g^{\alpha }$ is the current strength of the slip system. The hardening law for $g^{\alpha }$ and $\chi^{\alpha }$ can be obtained by:(5)$${\dot{g}}^{\alpha }=\sum_{\beta }h^{\alpha \beta }\left|{\dot{\gamma }}^{\beta }\right|$$
(6)$${\dot{\chi }}^{\alpha }=c{\dot{\gamma }}^{\alpha }-d\chi^{\alpha }\left|{\dot{\gamma }}^{\alpha }\right|$$

c and d are material constants; the hardening modulus $h^{\alpha \beta }$ was hardened using Peirce and Asaro’s law of self-hardening [30]:(7)$$h^{\alpha \alpha }=h\left(\gamma \right)=h_{0}sech^{2}\left|\frac{h_{0}\gamma }{g_{s}-g_{0}}\right|$$
where $h_{0}$ is the initial hardening modulus, $g_{0}$ is the initial slip resistance, $g_{s}$ is the saturated slip resistance, and the cumulative shear strain γ for all slip systems can be obtained from the following:(8)$$\gamma =\sum_{\alpha }\int_{0}^{t}\left|{\dot{\gamma }}^{\alpha }\right|dt$$

The latent hardening modulus $h^{\alpha \beta }$ is calculated as the ratio q of latent hardening to self-hardening. All 12 slip systems of FCC are considered as active slip systems:(9)$$h^{\alpha \beta }=qh\left(\gamma \right),\;\;for\;\alpha \neq \beta$$

In order to describe the FCI behavior of materials under cyclic loading, strain energy dissipation FIP is incorporated within the framework of classical crystal plasticity [18]. FCI occurs when the energy dissipation exceeds a critical threshold that remains consistent for a specific material. Strain energy dissipation $E_{p}$ can be expressed as:(10)$$E_{p}=\sum_{\alpha }\int_{0}^{t}{\dot{\gamma }}^{\alpha }\tau^{\alpha }dt=\sum_{\alpha }\int_{0}^{t}{\dot{\gamma }}^{\alpha }\sigma ∶\left(m^{\alpha }⊗n^{\alpha }\right)dt$$

### 2.3. Material Parameters

To simulate the elastoplastic behavior, the uniaxial tensile (strain rate of 0.05/min) and cyclic (symmetric strain amplitude of 1% at a frequency of 0.5 Hz) experimental data of Al-7.02Mg-1.98Zn (Peak aging state) [22] were fitted to Model A using RVE with a random orientation distribution. The polar figures of the orientation distribution along the <111> direction are presented in Figure 3. The findings are presented in Figure 4, where the results of the cyclic experiments were selected for ten cycles of data after cyclic stabilization. The crystal plasticity parameters fitted to the final Model A are listed in Table 1, and this parameter set was used for the intracrystalline regions of Models B and C.

The strength of age-hardened aluminum alloys primarily arises from solid solution strengthening and precipitation hardening. In contrast, the PFZ lacks both precipitates and solute atom depletion zones, resulting in a strength comparable to that of pure aluminum. Consequently, many scholars combine all factors other than solid solution and precipitation strengthening—such as the minor strength effects associated with micropores introduced during casting and residual stresses—into the matrix strength of aluminum, which is generally estimated to be around 50 MPa [31]. The work of Khadyko et al. [32] showed that $g_{0}$ and yield stress $\sigma_{y}$ were connected through the Taylor factor M: $g_{0}=\sigma_{y}/M$, where M is a value closely related to the statistical characteristics of the crystal orientation. The random orientation distribution is chosen in this work, so here M is equal to 3.067 [33]. Assuming that the PFZ and the intracrystalline region exhibit the same degree of hardening, denoted as ($g_{s}-g_{0}$), the crystal plasticity parameters for the PFZ can be derived as presented in Table 1.

### 2.4. Simulations

The three types of models are assigned corresponding material parameters, and random orientation distributions are incorporated. Models A and B share the same orientation distribution, which corresponds to that of the intracrystalline region in Model C. The PFZ orientation in Model C is assigned four distinct orientation distributions based on RF, and the polar figures of these distributions along the <111> direction are presented in Figure 3.

Two edges of the model were fixed, and a displacement load in the X direction was applied at one edge, as illustrated in Figure 5. Uniaxial tension and cyclic simulations were conducted, with the cyclic simulations performed under symmetric strain loading. The maximum displacement was set at 1%, and the displacements varied along a sinusoidal curve with a period of 2 s for a total of 10 cycles.

## 3. Results

### 3.1. Uniaxial Tension

Uniaxial loading was applied to the six models, yielding their respective stress-strain responses, as illustrated in Figure 6. The figure indicates that the stress response of the model without PFZ is significantly higher than that of the model containing PFZ, with a yield strength difference of approximately 80.0 MPa and a more pronounced strain-hardening effect. Additionally, there is no significant difference in the macroscopic stress responses of the model that includes PFZ, where the PFZ orientation of Model B aligns with the intracrystalline region, corresponding to the case where the rotation factor is 0.

The stress distribution of each model under uniaxial loading conditions is shown in Figure 7. The stresses in the figure are concentrated along the direction of 45° from the loading direction and are mainly concentrated near the grain boundaries. It is worth noting that compared with Model A, the stress concentration areas of the other models are smaller, and the stress in the PFZ is significantly larger than in the other areas. In the models containing PFZ, as the RF increases, the stress concentration area is more aggregated, which is shown in the figure by the occurrence of extreme values of stress at very few locations, and the maximum stress values of the other models are larger compared to Model B (RF of 0).

### 3.2. Cyclic Loading

The cyclic stress-strain curves of all six models exhibited pronounced cyclic hardening, as illustrated in Figure 8. Similar to the uniaxial stress-strain curves, Model A, which lacks PFZ, demonstrates a higher stress response and reaches a cyclic steady state after four cycles, with a peak stress of 400.0 MPa at stabilization. In contrast, the macroscopic stress-strain curves of the models containing PFZ show no significant differences among them. However, they reach a cyclic steady state after three cycles, with a peak stress of approximately 330.0 MPa at stabilization.

In this work, strain energy dissipation is employed as FIP. For the same material, with FCI, when the energy dissipation reaches a critical threshold, it has been demonstrated that the strain energy dissipation FIP effectively accounts for the influences of both stress and strain, thereby providing higher accuracy in the assessment of strain fatigue [18]. Figure 9 shows that FIP varies roughly linearly with loading cycles. Model A exhibits a higher average FIP than the other models, reaching approximately 72.0 kJ/m^2^ after 10 cycles. The differences in the average FIP among the models containing PFZ are minimal and are only evident in the locally zoomed-in plots, which illustrate that as the rotation factor increases, the average FIP also rises.

Fatigue failure often results from stress-strain concentration, which leads to localized accumulation of plastic deformation that gradually evolves into a crack source, as shown in Figure 10, which presents a cloud view of the FIP distribution for the six models after 10 cycles. Similar to the stress distribution observed in uniaxial tension, the FIP is also greater at 45° from the loading direction. The figure clearly indicates that the FIP is concentrated in a narrower region in the models containing PFZ. Specifically, a portion of the PFZ is significantly larger than the average value, while the FIP concentration is more dispersed in Model A. In contrast to the variation in the macroscopic average FIP, the maximum FIP in Model A without PFZ is less than 600.0 kJ/m^2^, whereas the maximum FIP in the models with PFZ exceeds 1000.0 kJ/m^2^, with the highest value reaching 1500.0 kJ/m^2^. Consequently, the models with PFZ exhibit a greater driving force for FCI compared to those without PFZ. This finding also elucidates the unusually low fatigue strength relative to tensile strength observed in age-hardened aluminum alloys. In the models containing PFZ, a higher RF corresponds to a larger maximum FIP value, which is consistent with the observed macroscopic phenomena.

## 4. Discussion

The material studied in this work is Al-7.02Mg-1.78Zn alloy with a face-centered cubic (FCC) crystal structure whose plastic deformation is mainly controlled by slip deformation. In aluminum polycrystals, there are 12 slip systems (SS) [34]. The slip system driving force under unidirectional loading is highly dependent on the Schmid factor (SF), which can be calculated for the slip system as follows:(11)$$SF^{\alpha }=cos\phi^{\alpha }\cdot cos\lambda^{\alpha }=\frac{\left(m^{\alpha }\cdot f\right)\cdot \left(n^{\alpha }\cdot f\right)}{\left|m^{\alpha }\right|\left|n^{\alpha }\right||f^{2}|}$$
where $\phi^{\alpha }$ and $\lambda^{\alpha }$ represent the angle between the loading direction (f) with the slip direction and normal to the slip plane, respectively. According to the von Mises criterion [35], five SSs need to be activated simultaneously in FCC crystals to ensure deformation compatibility by calculating the average ($\bar{SF}=\sum_{\alpha =1}^{5}max\left(\left|SF^{\alpha }\right|\right)/5$) of the top five SFs in the 12 SS as a driver of deformation.

During uniaxial loading, due to the low strength of PFZ, the dislocations are easy to slip and accumulate at grain boundaries when subjected to loads and take the lead in plastic deformation, and thus the model containing PFZ has a lower yield strength than Model A in terms of macroscopic performance. Figure 11 presents a cloud plot illustrating the distribution of initial and post-deformation $\bar{SF}$ in Model A. It indicates that as deformation progresses, the $\bar{SF}$ also increases. A distinct band at 45° from the loading direction is observable within the grains, suggesting that the grains rotate toward the direction of greater $\bar{SF}$ during plastic deformation. Notably, the position at 45° along the loading direction is particularly pronounced. Zhu et al. [36] showed that during uniaxial loading, the strain concentration of the specimen was mainly distributed at an inclination of about 45° to the tensile direction. At this point, the interior of the rotating grain is also divided into soft and hard zones due to the difference in $\bar{SF}$, where the region along the 45° position of $\bar{SF}$ is larger, which is softer and can take up more plastic deformation during deformation, and therefore stresses build up along these positions. The difference is that in models containing PFZ, the deformation is mainly taken up by the PFZ due to its much lower strength, so in these models, the stress concentration is limited to a small region of the PFZ.

During plastic deformation, grains must achieve deformation compatibility with their surrounding grains. It has been shown that dislocation slip is transmissible between certain grains, and this transmissibility allows good deformation coordination between adjacent grains, which can effectively relieve grain boundary stress concentration [37,38,39]. If there are significant differences in the deformation of neighboring grains, this can lead to pronounced stress concentrations at the grain boundaries. As shown in Figure 7, there are three places marked in white, among which one and two have obvious stress concentration. From Figure 11, it can be found that the $\bar{SF}$ of the neighboring grains in these two places are quite different, and the orientation difference between the two grains is 43° and 48°, respectively, which makes it difficult to achieve strain compatibility and produces obvious stress concentration. At Mark 3, although the $\bar{SF}$ of the neighboring grains also differed greatly, the orientation difference between the two grains was 25°, and the deformation transfer was easier due to the smaller orientation difference. For example, although Grain 34 in Figure 11a has a small $\bar{SF}$, there is a clear $\bar{SF}$ band inside the grain, which makes the deformation driving force increase, so that there is no stress concentration at the grain boundary.

Under strain-controlled cyclic loading, the presence of soft PFZ in the model allows for significant plastic deformation, resulting in a stress response similar to that of uniaxial loading, which is considerably lower than that of the model without PFZ. Additionally, dislocation motion occurs within the narrow PFZ, leading to rapid entanglement and locking of dislocations in a localized area. This mechanism enables the model containing PFZ to quickly reach a cyclic steady state. As illustrated in Figure 12, the cloud view of strain distribution at the end of 10 cycles reveals that the strain concentration region closely resembles the FIP concentration region shown in Figure 10. Notably, there is a slight difference between the two in Model A. This discrepancy arises because strain energy dissipation considers both stress and strain. At a position 45° to the loading direction, where there is a substantial deformation driving force, the strain is large, and stress concentration due to deformation incompatibility occurs at certain grain boundaries. Consequently, the FIP increases, making this location a potential zone of FCI. In the model containing PFZ, the stress concentration in the intragranular region is alleviated. Given that the strength of the PFZ is lower, the FIP at the 45° position along the loading direction of PFZ is significantly larger, making it more susceptible to becoming the FCI zone in aluminum alloys. Some researchers have noted that a primary characteristic of fracture in high-strength aluminum alloys is grain-boundary fracture, which typically occurs at 45° from the tensile axis [40,41].

During plastic deformation, the PFZ primarily absorbs an amount of plastic strain, while the intragranular region remains in the elastic stage. The elastic grains exert strong constraints on the PFZ, resulting in increased stress triaxiality and severe stress concentration within the PFZ. This condition further promotes FCI in the PFZ. With the increase of the RF, the orientation difference between the intragranular and PFZ increases synchronously, leading to further stress concentration in the PFZ. As shown in Figure 10, the value of the maximum FIP varies greatly, and when the RF reaches 0.5, the maximum FIP is 1.3 times that when the RF is 0, which greatly contributes to the fatigue crack driving force. However, the above occurs only in a tiny region of the PFZ, and the increase of RF has little effect on the intracrystalline portion so that macroscopically, the change in the FIP of the model containing the PFZ is almost uniform. Regardless of the model, FIP varies roughly linearly with loading cycles. For the same material, the energy dissipation has a critical value $w_{crit}$, and the FCI life can be calculated from the energy dissipation $w_{cyc}$ per cycle as:(12)$$N=\frac{w_{crit}}{w_{cyc}}$$

The method demonstrated in this work, although simulated for Al-7.02Mg-1.98Zn alloy, can be generalized to almost all age-strengthened aluminum alloys. The introduction of PFZ into the crystal plasticity model not only allows for the strain concentration effect due to the low strength of PFZ in age-strengthened aluminum alloys to be taken into account but also allows for the observation of stress concentration due to intracrystalline inhomogeneous rotations generated during the forming process of the aluminum alloy, which provides a significant advantage for the study of deformation behaviors and the prediction of fatigue life in age-strengthened aluminum alloys. Future work will be devoted to the use of this modeling approach for more complex geometries and multi-axial stress states.

## 5. Conclusions

In this work, the effect of PFZ on mechanical properties in Al-Mg-Zn alloys is thoroughly investigated by modeling the characteristic microstructure of age-strengthened aluminum alloys and performing crystal plasticity finite element simulations. The main conclusions are as follows:

(1)The presence of PFZ significantly reduces the yield strength of the alloy and further promotes stress concentration, especially in the PFZ;(2)In the PFZ separation model, the stress concentration is further enhanced by the increase in the orientation difference between the PFZ and the grain interior, thereby increasing the driving force for fatigue crack initiation;(3)Both stress and FIP converge along 45° from the loading direction, especially in the PFZ, indicating that age-strengthened aluminum alloys are characterized by cracking along grain boundaries;(4)PFZ is the key structure that undermines the fatigue performance of age-strengthened aluminum alloys. The fatigue crack driving force of the model including PFZ is more than 1.5 times higher than that of the model without PFZ, and the maximum FIP increases with increasing RF. Therefore, the fatigue life of the model without PFZ is much higher than that of the model with PFZ, which explains the lower fatigue performance of age-strengthened aluminum alloys than that of non-age-strengthened aluminum alloys.

These conclusions not only improve the understanding of the fatigue behavior of Al-Mg-Zn alloys but also provide important guidance for the design of higher-performance lightweight materials. Indeed, minimizing or even eliminating the presence of PFZ in age-strengthened aluminum alloys can significantly improve the mechanical properties. Additionally, reducing deformation non-uniformity during alloy forming can mitigate the stress concentration in the PFZ by reducing the orientation discrepancy between the PFZs.

## Figures and Tables

**Figure 1 materials-17-05623-f001:**
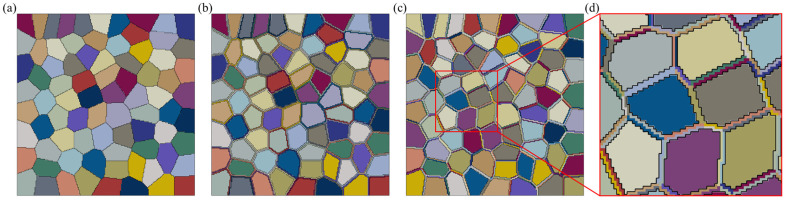
Schematic diagram of three types of polycrystalline geometry models: (**a**) model without PFZ; (**b**) model with complete PFZ per grain; (**c**) model that separates the PFZ into multiple regions; (**d**) local enlargement in (**c**).

**Figure 2 materials-17-05623-f002:**
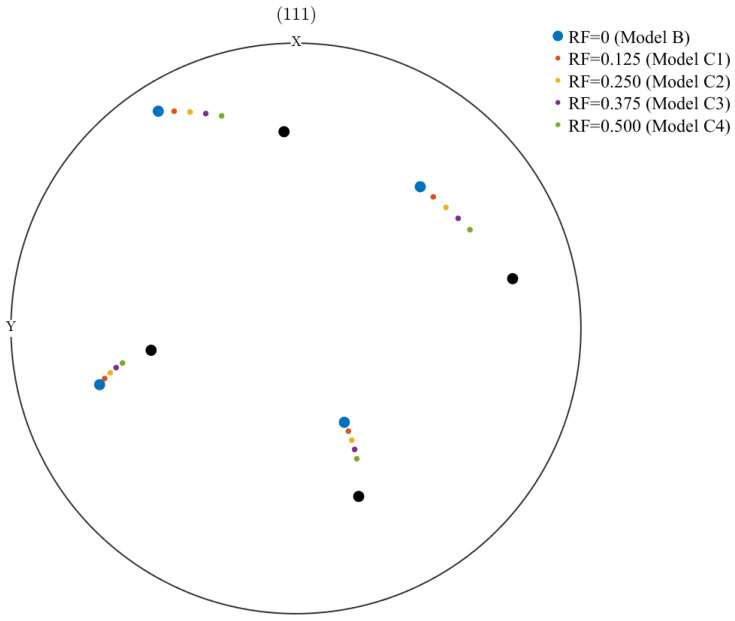
Polar figures of the orientation of the PFZ determined by the different rotation factors in Model C along the (111) direction.

**Figure 3 materials-17-05623-f003:**
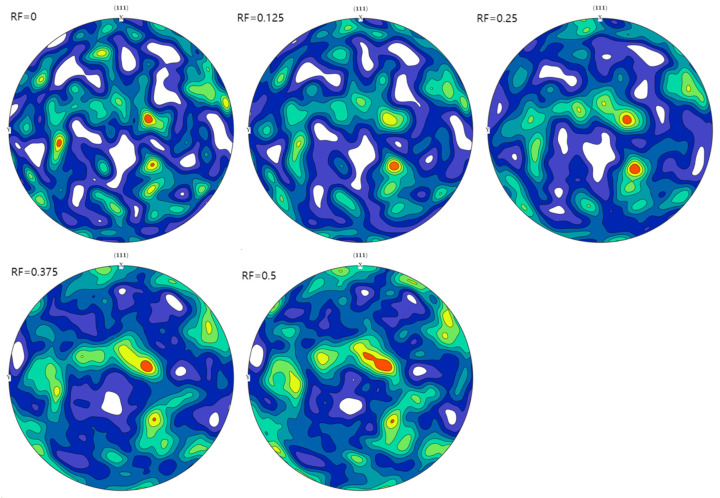
Polar figures along the <111> direction for models corresponding to different rotation factors in this work.

**Figure 4 materials-17-05623-f004:**
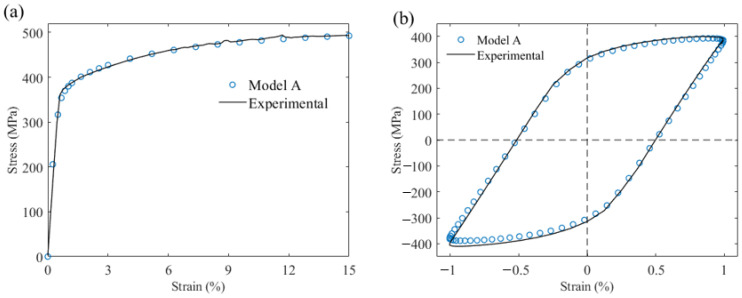
Comparison between simulation and experimental results for (**a**) tensile testing and (**b**) symmetric strain cycle testing.

**Figure 5 materials-17-05623-f005:**
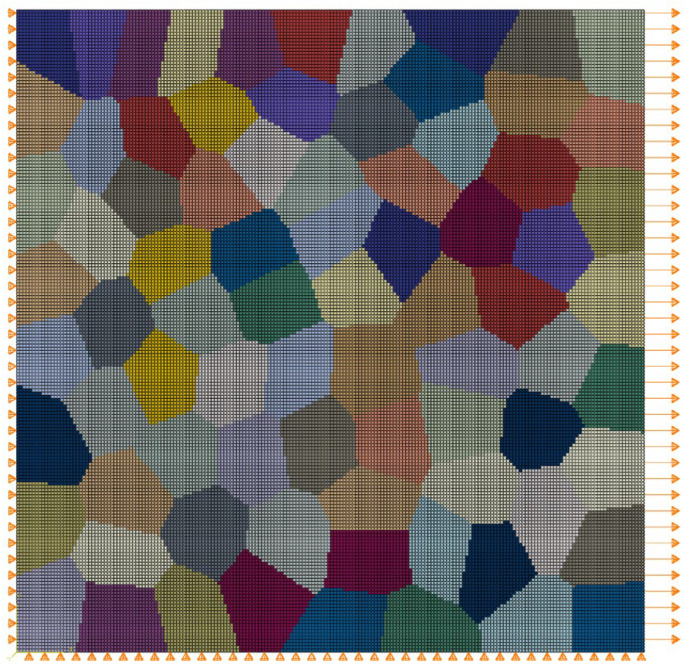
Schematic diagram of model loading in this work.

**Figure 6 materials-17-05623-f006:**
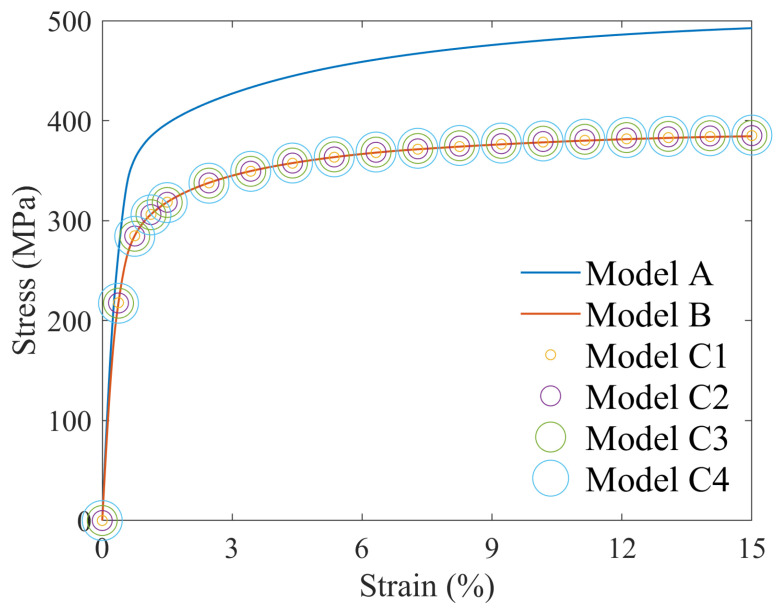
Stress-strain curves of models under uniaxial loading.

**Figure 7 materials-17-05623-f007:**
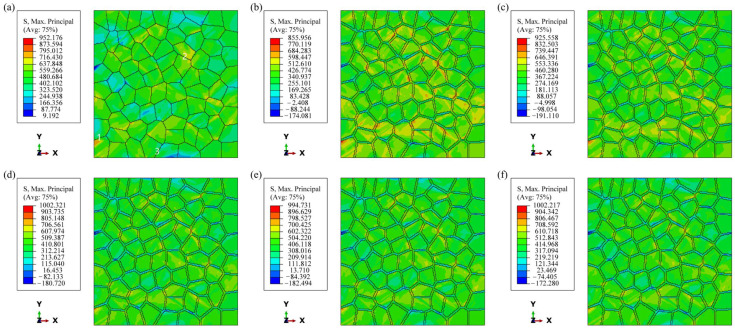
Stress distribution under uniaxial loading: (**a**) Model A; (**b**) Model B; (**c**) Model C1; (**d**) Model C2; (**e**) Model C3; (**f**) Model C4.

**Figure 8 materials-17-05623-f008:**
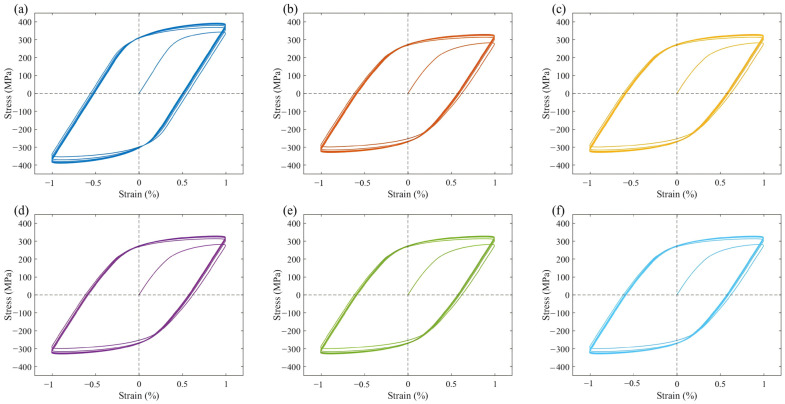
Stress-strain curves under cyclic loading for Models: (**a**) Model A; (**b**) Model B; (**c**) Model C1; (**d**) Model C2; (**e**) Model C3; (**f**) Model C4.

**Figure 9 materials-17-05623-f009:**
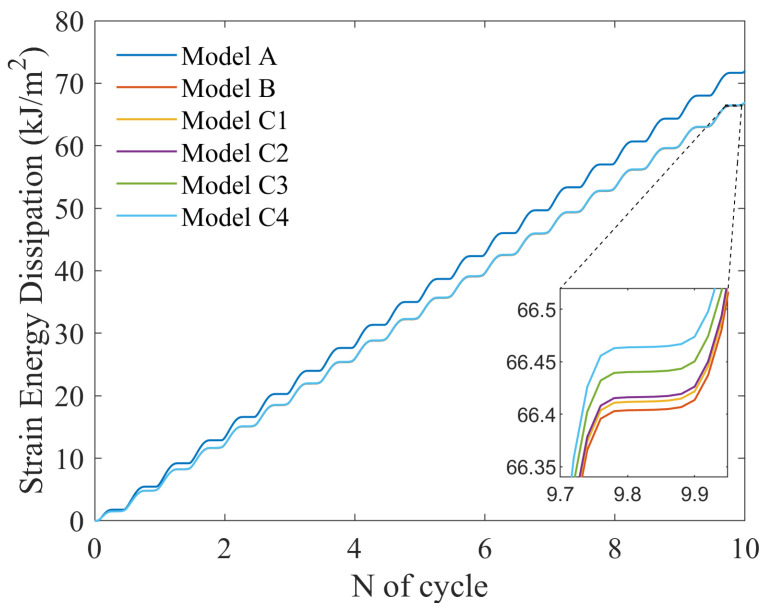
Variation of average strain energy dissipation with cycle time for models.

**Figure 10 materials-17-05623-f010:**
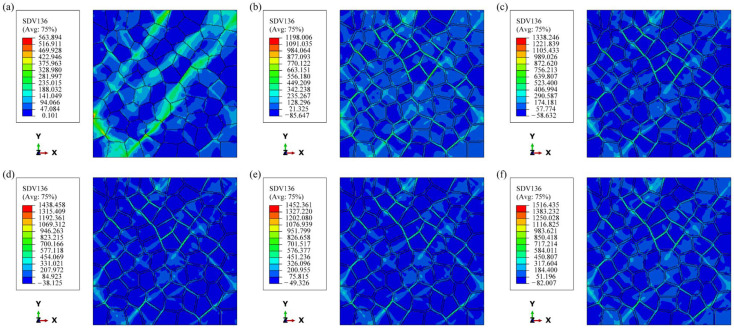
FIP distribution after 10 cycles: (**a**) Model A; (**b**) Model B; (**c**) Model C1; (**d**) Model C2; (**e**) Model C3; (**f**) Model C4.

**Figure 11 materials-17-05623-f011:**
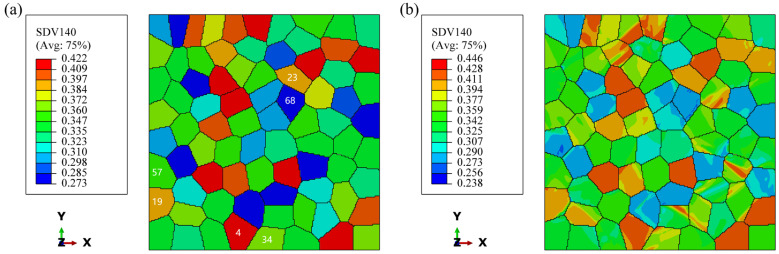
Distribution of $\bar{SF}$ in Model A: (**a**) before deformation; (**b**) after deformation.

**Figure 12 materials-17-05623-f012:**
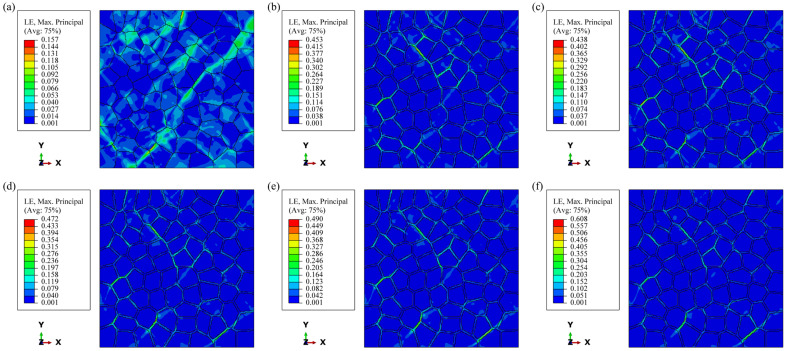
Strain distribution after 10 cycles: (**a**) Model A; (**b**) Model B; (**c**) Model C1; (**d**) Model C2; (**e**) Model C3; (**f**) Model C4.

**Table 1 materials-17-05623-t001:** Model parameters of intracrystalline and PFZ of Al-7.02Mg-1.78Zn used for simulations.

Elastic Parameters (GPa)				
$C_{11}$	$C_{12}$	$C_{44}$		
102.8	61.4	27.7		
Plastic Parameters				
${\dot{\gamma }}_{0}$ (s^−1^)	n	q	$h_{0}$ (MPa)	c (MPa)
0.001	50	1.4	65	350
$g_{0}^{int}$ (MPa)	$g_{s}^{int}$ (MPa)	$g_{0}^{PFZ}$ (MPa)	$g_{s}^{PFZ}$ (MPa)	d
115.0	130.0	16.6	31.6	25

## Data Availability

The data presented in this study are available on request from the corresponding author. The data are not publicly available due to privacy.

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
