# Peer review of "Effect of Precipitation-Free Zone on Fatigue Properties in Al-7.02Mg-1.98Zn Alloys: Crystal Plasticity Finite Element Analysis"

_materials, 2024, doi:10.3390/ma17225623_

Round 1
Reviewer 1 Report
Comments and Suggestions for Authors
1. The title of the article should be completed by indicating the group of aluminium alloys tested (Al-Mg-Zn), which corresponds to the scope and results of the research presented in the paper.
2. The abstract of the paper needs to be rewritten, clearly stating the scope of the theoretical and experimental studies performed and mentioning the specific aluminium alloy that was tested.
3. The paper contains the results of the theoretical research. On what basis did the authors show in Figures 4a and 4b the results of experimental tests for the tested alloy, if the authors did not present the methodology of the experimental tests, the type of specimens used for the individual tests and the test apparatus used. They did not state what guided the selection of specific parameters of the annealing process for the samples of the alloy under study.
4. The paper lacks any experimental verification of the theoretical results obtained for models A, B, C1, C2, C3 and C4.
5. In line 102 there is no reference to the literature [31].
6. In line 120, the authors should label the following figures with the letters: a), b), c) and d) and expand the caption under figure 1 accordingly.
7. In the conclusions, there is no statement of the influence of the areas without PFZ separations on the fatigue properties of the investigated alloy, which the authors write about in the title of the article.
8. In conclusion, there is no recommendation by the authors as to the practical use of the obtained research results in the manufacturing of lightweight, age-strengthened Al-Mg-Zn alloys, and intended for modern industries.
Author Response
Comments 1: The title of the article should be completed by indicating the group of aluminium alloys tested (Al-Mg-Zn), which corresponds to the scope and results of the research presented in the paper.
Response 1: Thank you for your valuable comment. We have changed the title.
Comments 2: The abstract of the paper needs to be rewritten, clearly stating the scope of the theoretical and experimental studies performed and mentioning the specific aluminium alloy that was tested.
Response 2: Thank you for your insightful comment, we have rewritten the abstract, please see lines 13-26 [Age-strengthened aluminum alloys, as important lightweight structural materials, have significantly lower fatigue properties than non-age-strengthened aluminum alloys. In this study, the polycrystalline models containing precipitation-free zones (PFZ) were constructed by secondary development of the traditional polycrystalline model by modifying the mesh file. Polycrystalline finite element simulations of peak age-treated Al-7.02Mg-1.98Zn alloys were carried out with this model. The results demonstrate that the PFZ's presence markedly reduces the alloy's yield strength and a substantial stress concentration occurs adjacent to the PFZ, generating significant compressive stresses at the PFZ. Under cyclic loading, the maximum strain energy dissipation in the model containing the PFZ far exceeds that observed in the conventional polycrystalline model, and the strain energy dissipation observed in the PFZ is significantly higher than that at other locations. This indicates that the PFZ is the main region for fatigue crack initiation. In addition, the introduction of a rotation factor to simulate the inhomogeneous rotation within the grain reveals that the additional stress concentration in the PFZ introduced by the aluminum alloy forming process further increases the fatigue crack initiation driving force.].
Comments 3: The paper contains the results of the theoretical research. On what basis did the authors show in Figures 4a and 4b the results of experimental tests for the tested alloy, if the authors did not present the methodology of the experimental tests, the type of specimens used for the individual tests and the test apparatus used. They did not state what guided the selection of specific parameters of the annealing process for the samples of the alloy under study.
Response 3: Thank you for your comment. This study is mainly focused on the methodology and therefore the experimental results have been used directly. We have added relevant experiments on uniaxial and cyclic loading to the manuscript, see lines 177-180 [To simulate the elastoplastic behavior, the uniaxial tensile (strain rate of 0.05/min) and cyclic (symmetric strain amplitude of 1% at a frequency of 0.5 Hz) experimental data of Al-7.02Mg-1.98Zn (Peak aging state) [31] were fitted to model A using RVE with a random orientation distribution.]. In this work, a precipitation-strengthened Al-7.02Mg-1.78Zn alloy, containing 90.78 wt% Al, 7.02 wt% Mg, 1.78 wt% Zn, and 0.1 wt% Zr, was casted in air and then homogenized at 470 ℃ for 24 h. After that, the alloy was extrusion deformed at 420 ℃ in a die with a diameter of 180 mm at a speed of 0.4 mm/s. The extrusion ratio is 11.5 and the exit temperature for the extrusion die was 395 ℃. The final dimensions of the extruded strips were 86 × 24.5 mm. Finally, the extruded strips were solution treated at 470 ℃ for 1 h, followed by quenching in water to room temperature. Subsequently, it was subjected to a two-stage aging process: the first stage at 90 ℃ for 24 h and the second at 140 ℃ for 24 h. The temperature and time of its peak aging were obtained by adjusting the temperature and time tested for a complete age-hardening curve.
Comments 4: The paper lacks any experimental verification of the theoretical results obtained for models A, B, C1, C2, C3 and C4.
Response 4: Thank you for your valuable comment. The focus of this work is on method demonstration and theoretical analysis, presenting a methodology based on the modeling of special microstructures in age-strengthened aluminum alloys in crystalline plasticity finite element simulations, which explains the low fatigue performance of this alloy. Namely, the model containing PFZ has better predictive performance than the conventional model for the micro-deformation and fatigue performance of this alloy. The present work does have shortcomings in terms of experimentation, which will be further verified in future work.
Comments 5: In line 102 there is no reference to the literature [31].
Response 5: Thank you for your professional suggestion. We have added citations and updated the manuscript to change the reference [31] to [22].
Comments 6: In line 120, the authors should label the following figures with the letters: a), b), c) and d) and expand the caption under figure 1 accordingly.
Response 6: Thank you for your comment. We have modified the content accordingly.
Comments 7: In the conclusions, there is no statement of the influence of the areas without PFZ separations on the fatigue properties of the investigated alloy, which the authors write about in the title of the article.
Response 7: Thank you for your comment. In the conclusions, we represent the case of PFZ separation by a rotation factor and describe the lower fatigue crack driving force of the model without PFZ. However, after your reminder, we discovered that this was not expressed clearly enough and have revised the conclusion, see lines 380-382 [In the model with PFZ separations, the stress concentration is further exacerbated by the increase in the orientation difference between the PFZ and the grain interior, thus increasing the driving force for fatigue crack initiation;] and 389-392 [Therefore, the fatigue life of the model without PFZ is much higher than that of the model with PFZ, which explains the lower fatigue performance of age-strengthened aluminum alloys than that of non-age-strengthened aluminum alloys.].
Comments 8: In conclusion, there is no recommendation by the authors as to the practical use of the obtained research results in the manufacturing of lightweight, age-strengthened Al-Mg-Zn alloys, and intended for modern industries.
Response 8: Thank you for your comments, relevant statements have been added to the conclusions based on the findings of this work, please see lines 393-399 [These conclusions not only improve the understanding of the fatigue behavior of Al-Mg-Zn alloys but also provide important guidance for the design of higher performance lightweight materials. Namely, minimizing or even eliminating the presence of PFZ in age-strengthened aluminum alloys can greatly enhance the mechanical properties. Additionally, the reduction of deformation non-uniformity during alloy forming can mitigate the stress concentration in the PFZ by reducing the orientation discrepancy between the PFZs.].
Reviewer 2 Report
Comments and Suggestions for Authors
Dear Authors,
I would like to thank you for the opportunity to review your article.
The research has a good scientific contribution. However, there is important information that needs to be adjusted for the publication of your article.
Please check the following points:
1. The abstract compares fatigue resistance with steels, indicating that aluminum has lower fatigue resistance. The correlation is not the most appropriate. The authors could use the correlation of aluminum alloys ASTM 2014, ASTM 7075, which are hardened by the T6 cycle (solutionization and artificial aging), and compare it with other alloys hardened by plastic deformation (hardening). What is the mechanism of movement of atomic dislocations in the precipitate-free zone? In the general context, compressive stresses, originated in the mechanical conformation of metals, favor the nucleation and propagation of cracks. The authors state the opposite in the abstract. Please justify. It is recommended that the abstract be rewritten, indicating the main knowledge gaps to be addressed, the methodology used, and the results in an objective manner.
2. In the introduction. What is the advantage of using the modified 5th series alloy with the addition of Zn, compared to the ASTM 2014 -T6, ASTM 6061 - T6 and ASTM 7075 - T6 alloys? Does this alloy have a different behavior? Did the modeling consider the alloy's solubilization conditions and the different artificial aging curves? Please justify. What are the maximum stress and yield stress considered in the simulation?
3. The methodology did not inform whether the material is solubilized for dissolving intermetallics. How were the aging temperatures and time determined? These data contribute to the simulation having a better basis in relation to the mechanical properties. Are there shorter aging cycles for this class of alloy? Did the authors find research on the Gleeble thermomechanical simulator used to determine the aging curves of this alloy?
4. Please improve discussions with other authors?
5. It is recommended that the conclusions be adjusted to the proposed objectives, correlating a conclusion to each objective or knowledge gap.
6. It is recommended that new articles with a high impact factor be added to the references to improve discussions with new authors.
Sincerely,
Author Response
Comments 1: The abstract compares fatigue resistance with steels, indicating that aluminum has lower fatigue resistance. The correlation is not the most appropriate. The authors could use the correlation of aluminum alloys ASTM 2014, ASTM 7075, which are hardened by the T6 cycle (solutionization and artificial aging), and compare it with other alloys hardened by plastic deformation (hardening). What is the mechanism of movement of atomic dislocations in the precipitate-free zone? In the general context, compressive stresses, originated in the mechanical conformation of metals, favor the nucleation and propagation of cracks. The authors state the opposite in the abstract. Please justify. It is recommended that the abstract be rewritten, indicating the main knowledge gaps to be addressed, the methodology used, and the results in an objective manner.
Response 1: Thank you for your insightful comment. We agree with you that this is an important consideration. We have removed the comparison with steel in the summary and modified it to a non-age-strengthened aluminum alloy. This is shown as [AA2014-T6 has a tensile strength of 490 MPa and a fatigue strength of 130 MPa, with a ratio of about 0.27; AA7075-T6 has a tensile strength of 560 MPa and a fatigue strength of 160 MPa, with a ratio of about 0.29; AA5083-H321 has a tensile strength of 330 MPa and a fatigue strength of 160 MPa, with a ratio of about 0.48; AA5049-H32 has a tensile strength of 240 MPa, fatigue strength of 120 MPa, and a ratio of about 0.5]. The movement of dislocations in the unprecipitated zone is also based on dislocation slip, and unlike other regions within the crystal, the movement of dislocations creates vacancies that can facilitate dynamic precipitation. The PFZ is depleted of solute atoms; therefore, dynamic precipitation cannot occur, only dislocation accumulation. Thanks to your reminder that compressive stress favors crack nucleation and propagation of cracks, we rechecked the simulation and found that the stress concentration area is near the PFZ, not inside the PFZ and that it is a state of compressive stress inside the PFZ, as shown in Fig. 7. Anyway, we have rewritten the abstract, please see lines 13-26 [Age-strengthened aluminum alloys, as important lightweight structural materials, have significantly lower fatigue properties than non-age-strengthened aluminum alloys. In this study, the polycrystalline models containing precipitation-free zones (PFZ) were constructed by secondary development of the traditional polycrystalline model by modifying the mesh file. Polycrystalline finite element simulations of peak age-treated Al-7.02Mg-1.98Zn alloys were carried out with this model. The results demonstrate that the PFZ's presence markedly reduces the alloy's yield strength and a substantial stress concentration occurs adjacent to the PFZ, generating significant compressive stresses at the PFZ. Under cyclic loading, the maximum strain energy dissipation in the model containing the PFZ far exceeds that observed in the conventional polycrystalline model, and the strain energy dissipation observed in the PFZ is significantly higher than that at other locations. This indicates that the PFZ is the main region for fatigue crack initiation. In addition, the introduction of a rotation factor to simulate the inhomogeneous rotation within the grain reveals that the additional stress concentration in the PFZ introduced by the aluminum alloy forming process further increases the fatigue crack initiation driving force.]. In addition, the introduction section has been amended accordingly, see lines 64-65 [n non-age-strengthened aluminum alloys the ratio is also about 1/2.].
Comments 2: In the introduction. What is the advantage of using the modified 5th series alloy with the addition of Zn, compared to the ASTM 2014 -T6, ASTM 6061 - T6 and ASTM 7075 - T6 alloys? Does this alloy have a different behavior? Did the modeling consider the alloy's solubilization conditions and the different artificial aging curves? Please justify. What are the maximum stress and yield stress considered in the simulation?
Response 2: Thank you for your valuable comment. In the introduction we give a brief description of the behavior of this alloy, see lines 43-47 [By adding Zn to the conventional 5xxx aluminum alloy, this new alloy can precipitate a large number of fine and dispersed nano T-Mg32(Al,Zn)49 phases through artificial aging, while maintaining the excellent corrosion resistance and good weldability characteristic of 5xxx aluminum alloys. The precipitated phase significantly enhances the alloy's strength through aging, resulting in improved mechanical properties and promising potential for applications in aerospace, transportation, and other fields [10–12].]. Other age-strengthened aluminum alloys, such as 2xxx with high strength, medium weldability, and very poor corrosion resistance; 6xxx aluminum alloys with average strength; and 7xxx aluminum alloys are commonly labeled as having poor corrosion resistance and weldability. After adding Zn modified Al-Mg-Zn alloy, the strength can reach 500MPa, and even some components of the strength up to 600MPa, and maintain good corrosion resistance and weldability. This work focuses on the methodology of PFZ impacts, and therefore only a brief representation of specific alloying behaviors is given. In the modeling, it is the state of peak aging that is directly considered and the material parameters are calibrated with the mechanical behavior of peak aging. The modeling of the different aging stages will be carried out in a subsequent work. In this simulation, instead of using stress as a loading condition, displacement loading is performed because displacement loading is more stable in finite element simulations. The maximum displacement in the simulation is chosen to be a value similar to the experimental case under peak aging, which can be seen in Figure 4.
Comments 3: The methodology did not inform whether the material is solubilized for dissolving intermetallics. How were the aging temperatures and time determined? These data contribute to the simulation having a better basis in relation to the mechanical properties. Are there shorter aging cycles for this class of alloy? Did the authors find research on the Gleeble thermomechanical simulator used to determine the aging curves of this alloy?
Response 3: Thank you for your comment. This study is mainly focused on the methodology and therefore the experimental results have been used directly. In practice, the alloys were prepared using pure Al (>99.9 wt.%), pure Zn (>99.95 wt.%), and pure Mg (>99.95 wt.%) as well as four intermediate alloys (Al-50% Cu, Al-5% Mn, Al-2% Sc and Al-5% Zr). The aging temperature was chosen to be double-stage aging, and since the small clusters precipitated under natural aging would revert under artificial aging, we chose a higher temperature for pre-aging, and we calibrated the pre-aging temperature by pre-aging for 24 hours at 80°C, 90°C, and 100°C, and then going through a peak aging at 140°C, and found that the best performance was achieved at 90°C, which led to the calibration of the pre-aging temperature. The secondary aging temperature was calibrated in the same way, and we chose 120°C, 140°C, 160°C, and 180°C all to reach peak aging, where 160°C and 180°C aging was correspondingly fast, but peaked at a lower value than 140°C, and 120°C aging response was too slow. As for the ageing time it was determined from the age-hardening curve. The aging regime of 90°C/24h+140°C/24h was finally determined. Yes, the Gleeble thermomechanical simulator is widely used for aging curve studies of alloys, but at the moment our experiments are not based on this simulation, and subsequently, we are considering incorporating this methodology into our research.
Comments 4: Please improve discussions with other authors?
Response 4: Thank you for your comment. We improve discussions with other authors to ensure that the manuscript is revised towards better quality.
Comments 5: It is recommended that the conclusions be adjusted to the proposed objectives, correlating a conclusion to each objective or knowledge gap.
Response 5: Thank you for your comment. The conclusions section has been modified based on our research, please see lines 380-382 [In the model with PFZ separations, the stress concentration is further exacerbated by the increase in the orientation difference between the PFZ and the grain interior, thus increasing the driving force for fatigue crack initiation;] and 389-392 [Therefore, the fatigue life of the model without PFZ is much higher than that of the model with PFZ, which explains the lower fatigue performance of age-strengthened aluminum alloys than that of non-age-strengthened aluminum alloys.].
Comments 6: It is recommended that new articles with a high impact factor be added to the references to improve discussions with new authors.
Response 6: Thank you for your insightful suggestion, we have added more references [37-40] to the discussion section, please see lines 304-306 [Zhu et al. [37] showed that during uniaxial loading, the strain concentration of the specimen was mainly distributed at an inclination of about 45° to the tensile direction.] and 316-318 [It has been shown that dislocation slip is transmissible between certain grains, and this transmissibility allows good deformation coordination between adjacent grains, which can effectively relieve grain boundary stress concentration [38–40].].
Round 2
Reviewer 2 Report
Comments and Suggestions for Authors
Dear Authors,
Most of the suggested corrections were applied and accepted.
However, it was found that the captions in Figure 2 needed to be adjusted. A general review of spelling and verbal agreement is also recommended.
Sincerely,
Author Response
Comments: Most of the suggested corrections were applied and accepted. However, it was found that the captions in Figure 2 needed to be adjusted. A general review of spelling and verbal agreement is also recommended.
Response: Thank you for your positive comments. We have changed the caption in Figure 2, see lines 136-137 [Figure 2. Polar figures of the orientation of the PFZ determined by the different rotation factors in model C along the (111) direction.], and increased font size in Figure 2. We have revised some of the spelling and verbal conventions in the manuscript, please see lines 14, 35, 39, 43, 45-46, 49, 52, 56, 60, 64, 66, 71, 78, 84, 86, 88, 92, 93, 96, 99, 386, 390, 392, 401, 402, 403.